# Driving Factors for R&D Intensity: Evidence from Global and Income-Level Panels

Cristiana Tudor [1,*] and Robert Sova [2]

1 International Business and Economics Department, The Bucharest University of Economic Studies, 010374 Bucharest, Romania

2 Management Information Systems Department, The Bucharest University of Economic Studies, 010374 Bucharest, Romania; robert.sova@ase.ro

* Correspondence: cristiana.tudor@net.ase.ro

**Abstract:** Research and development (R&D) has long been recognized as an important component of sustainable development, with a key role in the combatting of climate change. Moreover, R&D activity is increasingly acknowledged as an important contributing factor to global post-pandemic economic recovery. However, little is known about the determinants of R&D intensity (the share of R&D expenditure in GDP) and countries have repeatedly missed their set targets for this indicator. This article tackles this issue for a global panel consisting of 62 countries over the period 2007–2015 by using a dynamic system-generalized method of moments (SYS-GMM) panel model to uncover driving factors for R&D intensity. We also perform investigations on two homogenous subpanels constructed based on the income level of sample countries (High-income, and Middle- and Low-income subpanels), which contributes to assuring the robustness of results, along with formal model diagnostics and employment of alternative explanatory variables. We mainly find that the number of researchers is the most important driving factor for R&D intensity. High-technology exports have a statistically significant effect on R&D intensity only in middle and low-income countries. Patents are conducive to R&D intensity only in the high-income panel. Trade-openness is a significant mitigating factor for R&D investments throughout the panels and model specifications. Policy implications of results are also discussed.

**Keywords:** R&D intensity; sustainable development; impact factors; high-technology exports; human capital; trade openness; renewable energy consumption; patents; dynamic panel data; system-GMM

## 1. Introduction

The COVID-19 pandemic quickly transformed into the deepest social and economic crisis since World War II [1], posing major challenges for global innovation systems, as it endangered key production and innovation capabilities [2]. However, despite foreseen financial pressures, especially for public R&D expenditures [3], R&D investment is projected to play a key role for sustainable economic growth in the post-pandemic era (the so-called "Great Reset" as per [4]), contributing to underpinning private sector growth and job creation [5]). Moreover, science and innovation are acknowledged as quintessential factors in the post-COVID recovery period for world economies to tackle the climate emergency, meet the UN's Sustainable Development Goals of the 2030 Agenda, spur the digital transformation and promote more democratic and inclusive societies [2,6].

In fact, the crucial role of innovation for economic development and growth, presumed by the endogenous growth theory, has been validated by a large body of literature, from paramount earlier works [7–12] to more recent research [13–18].

Moreover, innovation is a documented mitigating factor for pollution, and spending on R&D contributes to decreasing carbon emissions in different countries and over different time periods [19–23] and also to increase energy efficiency by reducing carbon intensity [24]

The above-documented relationships also emerge from our study sample; Figure 1 reflects that the increasing share of R&D expenditure in GDP is generally positively related to economic income, and generally negatively related to carbon intensity (implying a negative relationship with polluting emissions and a positive relationship with energy efficiency), at world level.

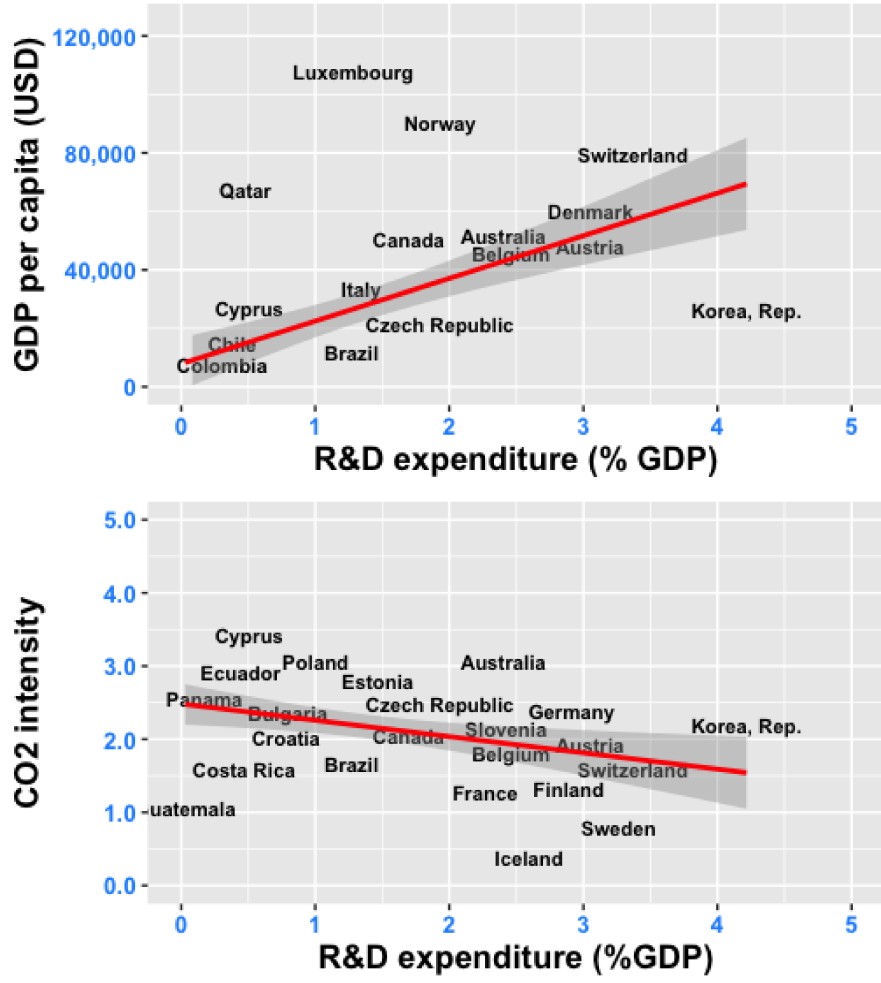

**Figure 1.** R&D expenditure (% GDP) and GDP per capita (USD), most recent year (Panel A); R&D expenditure (% GDP) and $CO_2$ intensity (kg per kg of oil equivalent energy use), most recent year (Panel B). Source of data: World Bank's Development Indicators (WDI) database.

Moreover, the world's top 10 investors in innovation reflected in Table 1 are all high-income countries and include amongst them world leaders in renewable energy, such as Sweden, Denmark, Germany or the US. The low-carbon technological innovation links, at the global level, innovative activities with the shift toward sustainable energy sources. Governments acknowledged the need to significantly increasing the public funding for technological innovation in low-carbon energy [25]. Moreover, sustainable energy technology represents around 80% of total public energy R&D spending, which in 2019 increased by 3% reaching USD 30 billion globally, while corporate energy R&D reached USD 90 billion [26].

**Table 1.** World top 10 investors in R&D, most recent year of available data per country (2015).

| Country | GDP Per Capita (Constant 2010 USD) | R&D Expenditures (% GDP) |
|---|---|---|
| Israel | 33,124 | 4.26 |
| Korea, Rep. | 26,064 | 4.22 |
| Japan | 47,103 | 3.28 |
| Sweden | 56,340 | 3.26 |
| Denmark | 60,402 | 3.05 |
| Austria | 47,789 | 3.05 |
| Germany | 45,208 | 2.91 |
| Finland | 45,648 | 2.89 |
| United States | 52,236 | 2.72 |
| Belgium | 45,507 | 2.46 |

Source of data: World Bank's Development Indicators (WDI) database.

As such, given its established role as a promoter of sustainable economic growth and contributor to mitigating pollution and tackling climate change at world level, with its acknowledged increasing importance in the post-pandemic era, our main research goal is to provide a deeper understanding of the determinants of countries' R&D investment, by examining its potential driving factors. Surprisingly, the related literature remains narrow, with few previous studies concerned with the determinants of R&D funding across countries, which further motivates our endeavor. Not in the least, it should be mentioned that countries, recognizing the importance of R&D investment for sustainable economic growth, have set minimum threshold levels for R&D intensity (i.e., the 3% goal set by the EU [27] or the 1% objective set by the African Union [28], which have been vastly unmet [6]. Moreover, ref. [29] concluded that countries should even reach R&D intensity levels as high as approximately 4 or 5% to achieve their INDC targets, numbers that not even world top investors in innovation reflected in Table 1 have managed to reach. Consequently, understanding driving factors for R&D intensity is paramount and has important policy implications at world level. All these factors are important motivators for this study.

Among the few related research, references [30,31] identifies population and GDP per capita income as key factors of R&D expenditure, whereas ref. [32] confirm that economic development (GDP per capita) is conducive to R&D investment. Ref. [33] finds that national culture and patent protection are factors that explain R&D investment, while also concluding that the degree of openness of an economy is unrelated to R&D intensity. In contrast, ref. [34] reveals that trade openness has a negative effect on domestic R&D, and that this effect decreases with an increase in GDP per capita and trade with OECD countries. Further, refs. [35,36] confirm the positive impact of intellectual property rights (IPRs) on innovation, whereas ref. [37] concludes that the investment in education quality leads to higher output of innovation activity.

This study contributes to the thin extant literature by investigating the impact of high-technology exports, number of researchers, renewable energy consumption, and trade openness on R&D intensity, defined as the share of R&D expenditure in GDP. Note that none of the previous studies has introduced this mix of explanatory variables. In robustness checks, we also use the number of patents as an alternative independent factor. Furthermore, unlike previous works that often use ordinary least squares (OLS), competing weighted least squares (WLS) or conventional panel analysis estimators, such as FE (fixed effects) and RE (random effects) to perform investigations, we rather employ a dynamic panel data model using the system-generalized method of moments (SYS-GMM) to undergo our investigation. This approach does not require distributional assumptions, and can allow for heteroscedasticity of unknown form references [38,39], thus solving serious

problems encountered in panel data estimations [40] (i.e., the endogeneity of regressors, the presence of fixed effects and autocorrelation within individuals) and bringing nontrivial efficiency gains [41]. Moreover, GMM is the method of estimation most suitable for panels with a small time dimension and a larger number of individuals [42] (as is the case in this study), allowing control for dynamic panel bias [43] and providing consistent estimates [44]. Consequently, based on the above arguments, we use the most robust method available to perform estimations.

Additionally, we make a further contribution by also investigating these linkages for two homogeneous groups of countries, constructed on the basis of the income (GDP per capita) level of sample countries, and on the basis of data availability: high-income (HI), and middle- and low-income (MLI) subpanels, respectively. This strategy thus permitted us to uncover any group specificity in terms of the impact that explanatory factors have on the dependent variable, represented by the R&D intensity, which in turn helped in identifying the most effective innovation policies.

Results for the global panel of 62 countries and for the two homogenous income-based subpanels contribute to the literature by providing empirical evidence for statistically significant drivers of R&D intensity, listed as follows. The R&D manpower (number of researchers) emerges as the most important driving factor for R&D intensity in all three panels, while high-technology exports have a statistically significant effect on R&D expenditure only for the middle and low-income panel. On the other hand, trade openness decreases R&D investments in all the panels, while renewable energy consumption does not significantly impact R&D funding in none of the panels. Feedback effects among variables and positive externalities are also uncovered in this study. Therefore, we find proof toward the effectiveness of innovation and economic policies that aim to increase R&D intensity by increasing the number of researchers involved in R&D activities. Moreover, the study concludes that less-developed countries should also implement policies that promote high-technology exports.

The remainder of this paper is organized as follows: Section 2 presents the data, offers an exploratory analysis of data, and explains the method and the overall conceptual framework of the investigation; Section 3 presents the estimation results together with a discussion of findings, and performs tests of results' robustness; finally, Section 4 contains the conclusions and policy implications of the study.

## 2. Materials and Methods

### 2.1. Data

Data for all variables employed in estimations were collected for the period 2007–2015 from the Word Bank's World Development Indicators (WDI). Depending on individual country/year data availability for the variables of interest (as per Table 2), we constructed an unbalanced panel covering a maximum period of 9 years for 62 countries. Thus, the resulting data panel emerged on the basis of data availability, where all countries with available data for the variables of interest for at least 3 years have been included.

Table 2 presents the variables employed in the empirical investigations, including abbreviations, WDI codes and variables' description.

**Table 2.** Variable description.

| Variable Abbreviation | Variable Code (World Bank WDI Database) | Variable Description |
|---|---|---|
| R&D Intensity | GB.XPD.RSDV.GD.ZS | Research and development expenditure (% of GDP) |
| HTEXP | TX.VAL.TECH.MF.ZS | High-technology exports * (% of manufactured exports) |
| NoR | SP.POP.SCIE.RD.P6 | Number of Researchers in R&D (per million people) |
| REC | EG.FEC.RNEW.ZS | Renewable energy consumption (% of total final energy consumption) |
| TradeOpen | NE.TRD.GNFS.ZS | Trade openness is the sum of exports and imports of goods and services measured as a share of gross domestic product (% of GDP) |

* According to the World's Bank definition, high-technology exports are "products with high R&D intensity, such as in aerospace, computers, pharmaceuticals, scientific instruments, and electrical machinery".

Our main variable of interest, R&D intensity, is represented by gross domestic expenditure on R&D, expressed as a percentage of GDP. According to the World's Bank definition, R&D expenditures include both capital and current expenditures in the four main sectors: business enterprise, government, higher education and private nonprofit, while covering basic research, applied research and experimental development. The rationale to select these factors is based on the main consideration that post-COVID policies will be influenced by the United Nations (UN) sustainability agenda. As such, both the dependent variable and potential explanatory factors for R&D intensity were mainly extracted from current policy position papers. Moreover, in establishing the mix of independent variables, we make the assumption that incorporating proxies of innovation inputs (i.e., number of researchers involved in R&D activities) and proxies of outputs (i.e., high-technology exports, renewable energy consumption) provides a much better understanding of the dependent variable by uncovering a potential process of incentivizing innovation through positive feedback, that is, a positive innovation impact of outputs would create an inventive to increase R&D intensity, which further contributes to increased R&D outlays. In robustness checks, we also employed another output of innovative activities in an economy, i.e., the number of patents as an alternative explanatory variable. Finally, trade openness was included in estimations not only for its use as a standard control variable in growth models ([45], but more importantly because in the context of technological innovation it can act as a carrier of knowledge spillovers by operating as a technology transfer channel [46–48]).

Further, the entire study sample is divided into two sub-samples of countries by income levels, which allows for the examination of potential asymmetric effects of drivers for R&D intensity according to different economic development stages. Thus: (1) the high-income (HI) group includes 38 countries with GDP per capita above USD 12500 (Australia, Austria, Belgium, Canada, Chile, Croatia, Cyprus, Czech Republic, Denmark, Estonia, Finland, France, Germany, Greece, Hungary, Iceland, Israel, Italy, Japan, Korea, Rep., Latvia, Lithuania, Luxembourg, Malta, Netherlands, New Zeeland, Norway, Poland, Portugal, Singapore, Slovak Republic, Slovenia, Spain, Sweden, Turkey, United Kingdom, United States, Uruguay); (2) the middle and low-income (ML) group includes 30 countries with a GDP per capita below USD 12,500 per capita (Armenia, Azerbaijan, Bosnia, Brazil, Bulgaria, Chile, China, Colombia, Costa Rica, Ecuador, Egypt, Arab Rep., Guatemala, Kazakhstan, Kyrgyz Republic, Latvia, Lithuania, Malaysia, Mexico, Moldova, North Macedonia, Peru, Poland, Romania, Russian Federation, South Africa, Thailand, Tunisia, Turkey, Ukraine, Uruguay). We should mention that some countries belong to more than one category over the whole sample period (i.e., Latvia, Lithuania, Poland, Uruguay, etc.), which is why the total number of unique countries included in the analysis ($N = 62$) is smaller than the sum of the two sub-samples lengths. We should also note that, due to data unavailability, a distinct low-income group of countries, similar to the World Bank's classification (with a GDP per capita below the threshold of USD 1000) could not be constructed with the study sample.

R software was used to implement the method and carry out estimations.

### 2.2. Exploratory Data Analysis

Some relevant exploratory tools of the data sample are the histograms of variables, shown in Figure 2, reflecting that all variables present right-skewed distributions, as most values cluster on the left. Interestingly, we notice that the distribution of R&D intensity is similar to the distribution of renewable energy consumption, which could be explained through the energy technological innovation channel that has been previously discussed.

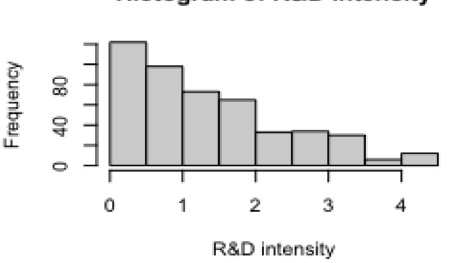
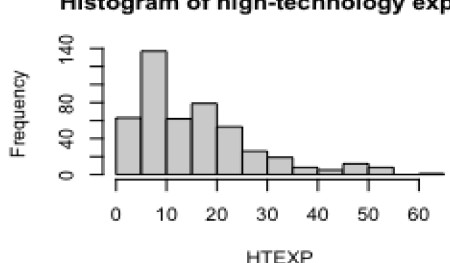

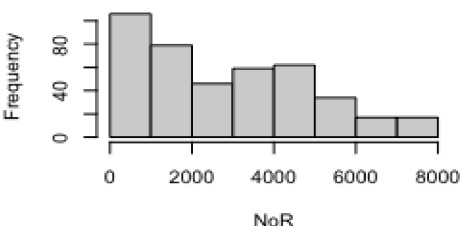
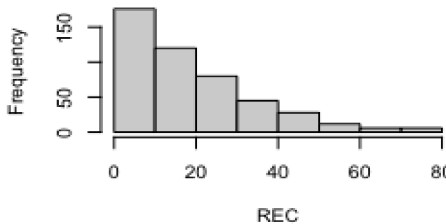

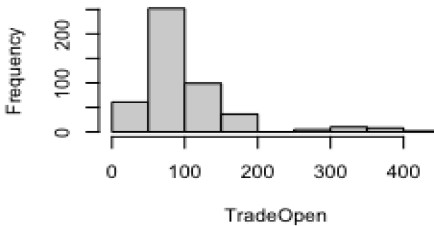

**Figure 2.** Histograms for the variables of interest.

Next, Table 3 presents the descriptive statistics for the full sample and the two sub-samples (high-, and middle- and low-income countries, respectively). We observed wide variations across subgroups. On average, high-income countries registered higher levels for all variables, and especially show much higher R&D intensity, and a significantly higher number of researchers. The renewable energy consumption is the only variable with small variations between the income samples.

Finally, Figure 3 reflects that there is a high heterogeneity across countries in what their investment in R&D is concerned, which should be taken into account for robust estimations. However, the global R&D expenditure in GDP remained stagnant through the analysis period, with only a slight increase in R&D funding registered in 2015.

**Table 3.** Descriptive statistics.

| Variable | Mean | Standard Deviation | Min | Max |
|---|---|---|---|---|
| Global panel | | | | |
| R&D Intensity | 1.37 | 1.06 | 0.02 | 4.43 |
| HTEXP | 15.43 | 11.70 | 0.53 | 60.71 |
| NoR | 2840.45 | 2038.51 | 17.38 | 7925.98 |
| REC | 18.73 | 15.91 | 0.20 | 77.12 |
| TradeOpen | 98.53 | 68.47 | 22.11 | 437.33 |
| High-income panel | | | | |
| R&D Intensity | 1.87 | 1.01 | 0.33 | 4.43 |
| HTEXP | 17.57 | 10.69 | 3.28 | 60.71 |
| NoR | 3808.10 | 1781.42 | 318.83 | 7925.98 |
| REC | 18.75 | 16.51 | 0.20 | 77.12 |
| TradeOpen | 111.50 | 80.43 | 24.49 | 437.33 |
| Low- and Middle-income panel | | | | |
| R&D Intensity | 0.55 | 0.40 | 0.02 | 2.03 |
| HTEXP | 11.88 | 12.45 | 0.53 | 50.87 |
| NoR | 925.75 | 759.65 | 17.38 | 3274.16 |
| REC | 18.69 | 14.91 | 1.16 | 67.44 |
| TradeOpen | 77.03 | 31.83 | 22.11 | 162.56 |

Source: Estimation results.

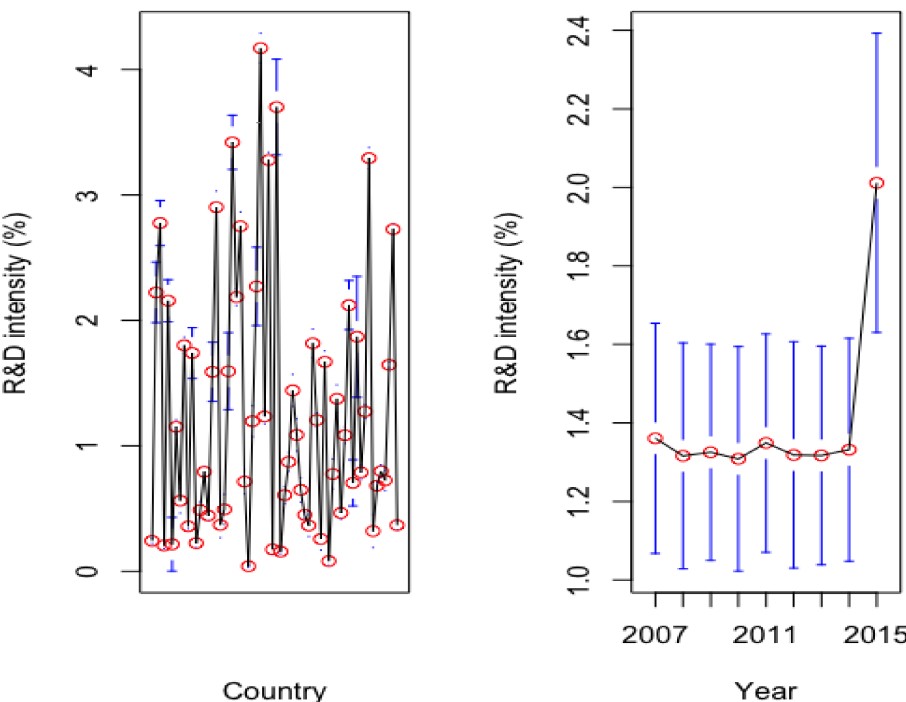

**Figure 3.** The mean level of R&D intensity by country, including the confidence intervals (Panel A). The evolution of mean R&D intensity at world level, with confidence intervals (Panel B). Source of data: World Bank's Development Indicators (WDI) database.

*2.3. Method*

The main relationship of interest that is based on the above discussion, and will be further investigated through a dynamic system-GMM panel model, is as follows:

$$\text{R\&D intensity} \sim \text{HTEXP} + \text{NoR} + \text{REC} + \text{TradeOpen} \tag{1}$$

where, as per Table 2, R&D intensity represents R&D expenditure (% of GDP) and it is a function of four variables including high-technology exports (HTEXP), number of researchers (NoR), renewable energy consumption (REC) and trade openness (TradeOpen).

Given the previous exploratory data analysis, in a manner similar to ref. [49], all the variables were converted to natural logarithm form before conducting estimations, which helps to smooth the data and produce more consistent results. Thus, Equation (1) can be rewritten in a log-linear form applied on panel data as follows:

$$LnR\&D\ Intensity_{it} = \beta_0 + \beta_1 LnHTEXP_{it} + \beta_2 LnNoR_{it} + \beta_3 LnREC_{it} + \beta_4 LnTradeOpen_{it} + \varepsilon_{it} \tag{2}$$

where $\beta_0$ designates the constant term; $\beta_1$, $\beta_2$, $\beta_3$ and $\beta_4$ are elasticities that represent the impacts of high-technology exports (HTEXP), number of researchers (NoR), renewable energy consumption (REC) and trade openness (TradeOpen) on R&D intensity; $\varepsilon$ denotes the error terms; the subscript $i$ ($i = 1, \dots, N$) denotes the country $i$ in the data sample, N = 62, and $t$ ($t = 1, \dots, T$) indicates the time period, with T = 9.

We estimated our dynamic panel data model using the generalized method of moments (GMM), which included the lagged level of R&D intensity. Moreover, in order to overcome the dynamic panel bias produced after the introduction of the lag of the dependent variable as an independent factor, we adopted the System GMM estimator, proposed by reference [50,51], which uses a set of instrumental variables that makes it robust in the presence of potential endogeneity of regressors [52,53] and/or heteroscedasticity and autocorrelation within individuals [42]. Thus, the System GMM estimator has been repeatedly validated as a strong estimator in numerous studies [43,53].

There are two types of GMM estimators (difference and system), which can both be employed either in a one-step or a two-step version. The set of instruments employed in estimations differentiates between the two estimators as follows: while the system-GMM estimator, which is employed in this research, also includes the lagged values of the dependent variable, the difference-GMM estimator solely takes into account all the available lags in difference of the endogenous and the strictly exogenous variables. Consequently, the system-GMM is superior as it permits dealing with neglected dynamics in static panel data models, resulting from ignoring the impact of lagged values of the dependent variable [40,54]. More details on the system GMM estimation and its advantages are found in ref. [55].

Consequently, the empirical model to be estimated will take the final form:

$$LnR\&D\ Intensity_{it} = \beta_0 + \beta_1 Ln(R\&D\ Intensity)_{it-1} + \beta_2 LnHTEXP_{it} + \beta_3 LnNoR_{it} + \beta_4 LnREC_{it}$$
$$+ \beta_5 LnTradeOpen_{it} + \mu_i + \phi_t + \varepsilon_{it} \tag{3}$$
$$i = 1, \dots, 62 \text{ and } t = 2007, \dots, 2015.$$

where the dependent variable representing R&D intensity is explained by its own lagged value and the four other explanatory variables included in Equation (3), respectively high-technology exports (HTEXP), number of researchers (NoR), renewable energy consumption (REC) and trade openness (TradeOpen), while $\mu_i$ stand for fixed country specific effects, $\phi_t$ represent time-effects and $\varepsilon_{it}$ is an error term with zero mean.

Given the structure of our panel data, which has a small time dimension (T = 9), we expected nonstationarity of variables to not be a concern, in particular as we estimated a dynamic model. However, panel unit root tests are helpful in identifying the right form in which the variables should be included in the empirical model, respectively their level or their growth form ref. [40].

Hence, the stationarity of series has been investigated with the CADF panel unit root test (Cross-Sectionally Augmented Dickey–Fuller) proposed by ref. [56], a second-generation unit root test that considers cross-sections dependence, which, as expected, rejected the null hypothesis of unit root and confirmed the stationarity of all variables in their levels (results are shown in Table 4).

**Table 4.** Results of CADF panel unit root tests.

| Variable | CADF Panel Unit Root Test |
|---|---|
| R&D Intensity | −6.2778 *** |
| HTEXP | −6.2644 *** |
| NoR | −4.978 *** |
| REC | −5.6731 *** |
| TradeOpen | −5.5304 *** |

*** Indicates significance at 1% level.

In order to ensure the consistency of the System GMM estimations, model diagnostics are further provided ([57]: firstly, the J-test of over-identifying restrictions of refs. [58,59] was calculated and reported, along with its *p*-values for the null hypothesis of instrument validity; secondly, we estimated the tests of ref. [60] for the first and the second-order serial correlations in the idiosyncratic remainder components or residuals [61]).

Figure 4 reflects the sequential steps taken for conducting this research, offering a clear overview of the implemented method.

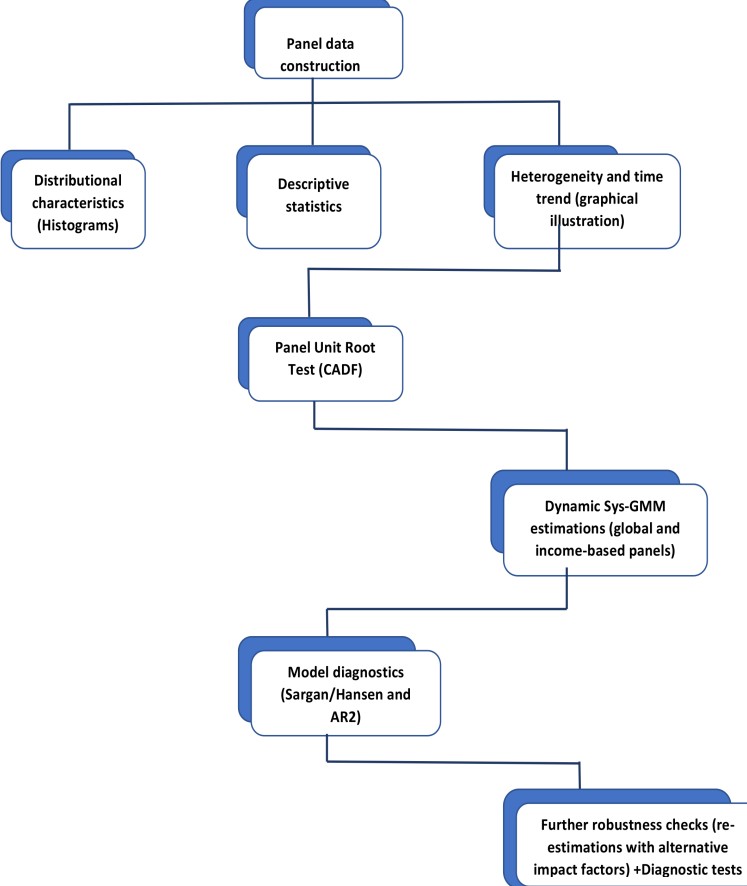

**Figure 4.** The sequential research steps.

## 3. Results and Discussion

The System GMM regression results from estimating Equation (3) over the period 2007–2015, for the global panel and the two sub-panels corresponding to the two income-groups of countries are reported in Table 5. We also checked the robustness of our results against alternative measures of R&D output (i.e., the number of patents), and these results are reported in Table 6. As the two-step GMM estimators can be seriously biased downwards in finite samples [51], similar to ref. [62] we preferred to employ the one-step version of the system GMM estimation for all panels and model specifications. Additionally, as mentioned earlier, the robustness of the System GMM estimators depends both on the assumption that the error term does not have a serial correlation problem and on the validity of instruments. Consequently, these assumptions were verified through the Arellano–Bond test for no serial correlation in the error terms and the Hansen/Sargan test for the validity of the instruments, which are reported in the bottom rows of Tables 5 and 6, respectively. According to ref. [60], the GMM estimator requires that there is no second-order serial correlation (AR2 test) in the differenced residuals. In all estimations in this study, these diagnostics support the validity of the model specifications. Note also that, in all specifications, the Hansen/Sargan test does not reject the null hypothesis of instrument validity, consistently indicating a high chance of a type one error if the null is rejected (for example, ref. [63] stated that even a *p*-value as high as 0.25 "should be viewed with concern"; our results indicate *p*-values of at least 0.6, which allows for a high degree of confidence in the validity of instruments). All diagnostic tests thus confirm that all System GMM equations are properly specified; thus, we can proceed with presenting and discussing the estimations' results.

The findings for all three panels confirm our a priori assumption, namely, that higher R&D intensity in the previous period contributes to higher R&D intensity in the current period. At a global level, a 1% increase in the lagged R&D intensity advanced the investment in R&D in the current period by about 0.40%, while the results for the income-based sub-panels show similar levels.

**Table 5.** Effect of explanatory variables on R&D intensity: one-step system-GMM dynamic panel estimation regression results for the global, high-income and low- and middle-income panels.

| | **Global** | **High-Income (HI)** | **Low- and Middle-Income (LMI)** |
|---|---|---|---|
| Dependent variable: R&D intensity | | | |
| Independent variables | Estimate | | |
| R&D Intensity (−1) | 0.400225 *** | 0.357990 *** | 0.3843145 *** |
| TradeOpen | −0.256184 *** | −0.139277 *** | −0.4507593 *** |
| NoR | 0.467285 *** | 0.589239 *** | 0.3931260 *** |
| HTEXP | 0.050262 | 0.063533 | 0.0923286 ** |
| REC | −0.019476 | −0.028049 | 0.0037797 |
| Hansen/Sargan J-test (*p*-value) | 0.8642 | 0.6387 | 0.7231 |
| AR2 test (*p*-value) | 0.74768 | 0.53016 | 0.45671 |

** Indicates significance at 5% level. *** Indicates significance at 1% level.

**Table 6.** Robustness checks: Effect of alternative explanatory variables on R&D intensity: one-step system-GMM dynamic panel estimation regression results for the global, high-income and low- and middle-income panels.

| | **Global** | **High-Income (HI)** | **Low- and Middle-Income (LMI)** |
|---|---|---|---|
| Dependent variable: R&D intensity | | | |
| Independent variables | Estimate | | |
| R&D Intensity (−1) | 0.3282568 *** | 0.262483 *** | 0.376961 *** |
| TradeOpen | −0.2634110 *** | 0.116984 | −0.297510 ** |
| NoR | 0.4977531 *** | 0.552330 *** | 0.338678 *** |
| Patent | 0.0097895 | 0.080565 *** | 0.043666 |
| REC | −0.0202989 | 0.041230 | 0.033483 |
| Hansen/Sargan J-test (*p*-value) | 0.7634 | 0.5946 | 0.6675 |
| AR2 test (*p*-value) | 0.7397 | 0.99914 | 0.4782 |

** Indicates significance at 5% level. *** Indicates significance at 1% level.

Furthermore, another significant (negative) impact, which is present throughout the panels, is found for the trade openness variable, such that a 1% increase in the trade openness decreases R&D expenditures worldwide by 0.25%, but the impact is significantly higher for low and middle-income countries (i.e., −0.45%) than for rich countries, where a 1% increase in openness decreases the investment in R&D by −0.14%. As such, it can be extracted that a greater trade openness, which permits the rapid transmission of knowledge and innovation, could be a good policy for beneficiating from innovation transfer, and consequently affects domestic R&D investment decisions. In this respect, our findings support previous studies [64] that concluded that technology spills across countries through the channel of trade flows. Moreover, the low- and middle-income countries seem to beneficiate more from knowledge and innovation spillover than their developed counterparts. Thus, our results confirm that openness is an important channel for promoting technological progress, supporting the findings of refs. [34,65]. The negative relationship between trade openness and R&D intensity also implies that innovation spillovers through the trade channel and R&D investment act as substitutes, rather than complements, for the sample of countries in this study. However, it should be noted that, in robustness checks (Table 6) innovation stops spilling through the trade channel in the case of high-income countries, when patents are introduced as explanatory variables. This is explained by the stronger intellectual property rights (IPRs) in this group of countries and further confirms the findings of ref. [66], which show that patent rights have a significant positive impact on the motivation to innovate in wealthy nations, while this effect is statistically insignificant in developing countries.

Furthermore, results indicate that the number of researchers involved in R&D is the main factor that increases R&D intensity in the investigated countries, worldwide and across the income-groups of countries. However, the relationship is stronger for high-income countries, where a 1% increase in the number of researchers increases R&D intensity by 0.58%, whereas for low and middle-income countries the impact is lower (i.e., 0.39%). Additionally, these results remain robust across different model specifications. Our findings are in line with those of ref. [67] that shows that the scientific researchers in a country are robust determinants for R&D intensity and confirm that highly skilled human resources are crucial for a country's research and innovation capacity and competitiveness [68]. The findings also validate the endogenous growth model proposed by ref. [69], whereby human capital is a key input factor in the R&D sector that generate new designs, which at their turn generate new investment opportunities and complement those of ref. [70] that conclude that human capital is the most significant driving factor for innovation, as proxied by current patent applications per capita.

Moreover, a disaggregation occurs between high-income and low- and middle-income countries whereby the factor representing high-technology exports has a significant positive effect on R&D intensity in the low and middle-income countries, while it has no significant impact on R&D intensity in the high-income countries. In fact, for countries included in the LMI panel, the increase in high-technology exports by 1% results in an increase in R&D intensity of 0.09%, whereas this propensity is not found in the case of other countries. We should recall that the UN's 2030 Agenda encourages technical advancement, particularly in developing countries, through research and innovation [71]. Additionally, the share of high-technology exports in total manufactured exports is one relevant indicator that reflects the progress made toward achieving that goal. The greater the proportion of exports in higher technological complexity categories, the more evidence that the economy's structural transformation has advanced. Moreover, as ref. [72] showed, R&D intensity enhances downstream commercialization and diffusion activities, such as increasing high-technology exports. Thus, our findings confirm that R&D and innovation are critical to this shift in the case of low and middle-income countries because they lay the groundwork for the adoption of new, more efficient technologies, which lead to increased high-technology exports that in turn are conducive to increased R&D funding. Consequently, we find proof of positive feedback between high-technology exports and R&D intensity, reflecting that developing countries are in the right track toward the achievement of the 9th sustainable development goal (SDG) of the 2030 Agenda. This effect is not present for high-income countries, attesting previous findings [3] that the gap between developing and developed countries has narrowed as developing economies have been catching up in the structural transformation of manufactured export. On the contrary, when we redid our estimations by using an alternative measure of R&D output (i.e., the number of patents instead of high-technology exports), the findings in Table 6 show that patents are conducive to R&D intensity only in developed countries, whereas this impact is not present for low- and middle-income economies. This further implies that a positive feedback effect between patents and R&D funding exists only in high-income countries, whereby an increase in innovation output (i.e., number of patents) determines higher R&D funding, which in turn stimulates innovative activities and the creation of more innovations. Furthermore, these data imply that the patent gap between developing and developed countries remains wide, whereas it has shrunk in terms of structural change of manufactured exports.

Finally, results show no statistically significant relationship between renewable energy consumption and R&D intensity for all three panels of countries, and thus find that, globally, an increase in renewable energy consumption is not conducive to increased investments in R&D. On the other hand, a reverse effect has been previously encountered for rich countries, whereas increased R&D funding has been showed to spur the consumption of renewable energy [73]. Thus, the world's top investors in innovation include amongst them world leaders in renewable energy [74,75]. However, we did not encounter a feedback effect between REC and R&D intensity. There may be several reasons for such a finding. One explanation for this result is that, although currently the cost of renewable energy registered dramatic falls [76,77], this was not the case over the analysis period in this study [78] and as such, over 2007-2015, countries did incur economic costs in their path toward carbon neutrality, which would have consequently eliminated the propensity to undertake new R&D investments. Additionally, this relationship could be explained by the fact that countries generally rely on innovation diffusion through the trade flow channel to increase their consumption of sustainable energy [40].

## 4. Conclusions

Governments and international agencies have long recognized that R&D and innovation contribute to sustainable economic development and play a key role in combatting climate change. Consequently, countries have set goals for minimum investment in R&D in proportion to GDP (i.e., R&D intensity), which have been vastly unmet. This incongruence between policy and reality is particularly worrisome, as R&D intensity has been acknowl-

edged as a crucial factor for the "Great Reset" in the aftermath of the COVID-19 pandemic. Consequently, understanding the driving factors for R&D intensity is a timely research topic, with important policy implications.

This study develops a dynamic panel data model by employing the generalized method of moments (System GMM) to estimate the impact of high-technology exports, number of researchers, renewable energy consumption and trade openness on R&D intensity, defined as the share of research and development expenditure in GDP. We assured the robustness of results by extracting the empirical evidence from a global unbalanced panel consisting of 62 countries, as well as from two income-based subpanels (high-income and middle- and low-income, respectively), from performing two models of diagnostic tests (i.e., the Arellano–Bond test for no serial correlation in the error terms and the Hansen/Sargan test for the validity of the instruments), and also through re-estimating the models for all three panels by employing an alternative measure of R&D outlays (i.e., the number of patents).

The main results over the period 2007–2015 indicate that: (i) human capital (R&D manpower) is the most important driving factor for R&D intensity in all the panels we considered, and it holds statistical significance in all model specifications; (ii) high-technology exports have a statistically significant effect on the research & development expenditure only for the middle- and low-income panel, confirming that developing economies have made progress in the structural transformation of manufactured exports; (iii) patents are a determinant of R&D intensity only in the high-income panel; (iv) trade openness mitigates R&D investments for all the panels, implying that innovation is disseminated through the trade openness channel; and (v). renewable energy consumption is not conducive to R&D intensity in none of the three panels of countries we consider, indicating that green technology innovation disseminates through the trade channel, rather than increasing through R&D investment.

These results have important policy implications. Policy makers must consider that a feedback effect is at work: human capital increases R&D intensity, which in turn spurs investment in innovative activities, and the latter bring forward the need for more R&D manpower. Moreover, it should be acknowledged that positive externalities also emerge, as R&D human capital promotes R&D intensity and the latter enhances economic development and growth. All these effects must be taken into account for the design of effective and efficient policy. Additionally, the issuers of policy, especially in the less-developed countries, should consider another feedback effect: high-technology exports are conducive to increased R&D funding, and the latter drives the adoption of new, more efficient technologies, which in turn lead to increased high-technology exports. For high-income countries, a similar feedback effect between the number of patents and R&D intensity should also be considered.

However, the pandemic-induced crisis has compelled world governments to direct significant resources to other priority areas. Consequently, as [79] warns, government financing for R&D is declining at a time when global concerns such as climate change and aging demographics deserve responses. The stakes are higher for low-income countries that currently have to juggle limited financial resources, pressure to answer current pandemic challenges that are absorbing a growing share of public resources and pressure to increase R&D funding.

Thus, we argue that countries should make use of post-pandemic stimulus and recovery packages to protect and enhance their innovation systems through implementing policies extracted from the main relationships that emerge from this study. Complementarily, middle- and low-income countries should also consider that trade openness is an important channel for promoting technological progress that acts as a substitute for R&D investments. As a result, this group of countries should take advantage of research findings suggesting that greater trade openness, which allows for the rapid transmission of knowledge and innovation, could be a good policy for benefiting from innovation transfer

and, as a result, reduce financial pressures caused by R&D funding in the aftermath of the global pandemic.

**Author Contributions:** Conceptualization, C.T.; methodology, C.T. and R.S.; software, C.T. and R.S.; validation, C.T.; formal analysis, C.T.; investigation, C.T.; data, C.T.; writing—original draft preparation, C.T.; visualization, C.T.; supervision, C.T. and R.S.; project administration, C.T. All authors have read and agreed to the published version of the manuscript.

**Funding:** This research received no external funding.

**Institutional Review Board Statement:** Not applicable.

**Informed Consent Statement:** Not applicable.

**Data Availability Statement:** Data is publicly available from the World Bank's Development Indicators (WDI) database.

**Conflicts of Interest:** The authors declare no conflict of interest.

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
