# Peer review of "Driving Factors for R&D Intensity: Evidence from Global and Income-Level Panels"

_sustainability, doi:10.3390/su14031854_

Round 1

Reviewer 1 Report

More emphasis on the financial pressure on countries need to be embedded

English language and style are fine/minor spell check required

Reviewer 2 Report

The research is interesting and the paper's structure shows a logical development and a good documentation.

The methodologies are well presented and appropriate  to the proposed objective of the research paper.

However, I consider necessary to review the line 283 and 306 because the equations require some editing.

It could be necessary to explain better the differences of the sign of coefficient in the case of renewable energy between high, middle -income countries and global. Why the positive impact is only in the case of middle-income countries? Also , why in the robustness checks, the sign of coefficients in the case of high-income countries are different in the case of REC and Trade Open variable as compared to Table 5. How does the absence of the HTEXP variable explain such differences?

I consider that the paper can be improved in the Conclusion section by differentiating the policy recommendations between group of countries.

Reviewer 3 Report

The empirical strategy needs clarification. Please justify using the dynamic system-generalized method of moments (SYS-GMM) panel model to uncover driving factors for R&D intensity. There is a discussion from lines number 117 to 126. However, these lines can be improved to justify using these estimation techniques to determine the driving factors for R&D intensity. 

The reader might struggle to arrive at the Research Objective and Research Questions. Would you please mention these critical components of your research paper (the Research Objective and Research Questions) clearly? 

Please explain the period selection since the WDI usually are available for a wide range of periods. 

Author Response

This manuscript is a resubmission of an earlier submission. The following is a list of the peer review reports and author responses from that submission.